

# The climate of Granada (southern Spain) during the first third of the 18th century (1706-1730) according to documentary sources.

Fernando. S. Rodrigo[1]

[1]Department of Chemistry and Physics, University of Almería (Spain)

*Correspondence to*: F.S. Rodrigo (frodrigo@ual.es)

**Abstract. T**he climatic information recorded by the physician Francisco Fernández Navarrete in Granada (southern Spain) during the first third of the 18th century is analysed in this work. Observations are included in the book *Cielo y suelo granadino* (*Sky and soil in Granada*), and consist of qualitative comments relating climatic conditions to illness and diseases since 1706, as well as instrumental observations (using an 'English barometer' and a 'Florentine thermometer') from December 1728 to
February 1730. To the best of our knowledge, these are the earliest instrumental observations recorded in Spain. The analysis shows that seasonal mean values of temperature and precipitation during the period 1706-1730 were very similar to those of periods of similar length at the beginning of the 20th century, as 1906-1930. However, some years were especially extreme, such as the dry first half of the 1720s decade, or the winter 1728-1729, when a strong cold wave affected the city.

## 1 Introduction.

Historical climatology offers the possibility of reconstructing climatic conditions during the pre-instrumental period, that is, before the establishment of meteorological observation networks around 1850. Documentary sources are basic data sources for this time period, because record climatic anomalies and extreme events, making it possible to relate such events to climatic changes. Last years a great amount of papers have been published using the methodological basis of historical climatology (Brázdil et al., 2005; 2010). In addition, the recovering of the early instrumental observations is a priority objective in climatic
research (Brönnimann et al., 2018).

There are many works on the historical climate in the Iberian Peninsula using documentary sources and early instrumental observations from Spain and Portugal (see, for instance, Bullón, 2008; Domínguez-Castro et al., 2010, 2014; Alcaforado et al., 2012; Fragoso et al., 2015). The first meteorological measurements in the Iberian Peninsula were taken in Portugal between 1 November 1724 and 11 January 1725 (Domínguez-Castro et al., 2013). In Spain normally has been considered that the
*Ephemerides barométrico-médicas matritenses* (*Ephemeris barometric-medical from Madrid)* by the physician Francisco Fernández Navarrete, was the first meteorological instrumental series (Anduaga Egaña, 2012). It is a set of daily and sub daily meteorological observations taken in Madrid between March and October 1737. In this work we present a set of observations taken by the same observer in Granada (to the south of the country) some years before, between December 1728 and February 1730. These observations are included in a handwritten book dated on 1732 and kept in the Archive of the Franciscan Order
in Cataluña (Gil Albarracín, 1997). The title of the book is *Cielo y suelo granadino* (*Sky and soil in Granada*), and it may be considered as one of the first Spanish medical treatises that followed the neo-Hippocratic hypothesis concerning the influence of climate on human health. In following sections these observations along with qualitative comments by the author on the climatic conditions since 1706 will be analyzed.

The climatic interest of Granada, to the south of the Iberian Peninsula, is due not only to its geographic location (latitude
37°10'N, longitude 3°36'W), near the Mediterranean Sea, exposed to Atlantic disturbances and Mediterranean influences, but also to its height, 660 meters above sea level, and proximity to the highest mountain ridge in the Iberian Peninsula, Sierra Nevada mountains, with some peaks 3000 meters above sea level (Fig.1). The study period is interesting because it begins at





the ending of the cold period called Maunder Minimum (Owens et al., 2017), and continues during subsequent decades. Therefore, it allows to explore the climate behavior in a city located in the Mediterranean area (hot-spot of climatic change (Giorgi, 2006)), when natural climatic changes occurred.

The scheme of the paper is as follows: biographical and bibliographical information on the author and his texts are described
in Section 2; Section 3 studies general conditions during the period 1706-1730, and section 4 is focused on the instrumental observations from December 1728 to February 1730; finally, some conclusion remarks are included in the last section.

**2 The observer: Francisco Fernández Navarrete.**

Francisco Fernández Navarrete (Granada, 1680; Madrid, 1742) studied medicine in Granada, where lived until 1734, when he moved to Madrid as doctor of the king Felipe V. He was an active member of the Royal Academy of Medicine (founded on
1734), and the Royal Academy of History (founded on 1738). He developed his work following the neo-Hippocratic hypothesis. According to this medical paradigm, illness, epidemics, and public health are related to environmental conditions, in particular to the variability of meteorological variables (Demareé, 1996). This idea was predominant in Spain until at least mid-19th century (Rodrigo, 2016). So, it is not surprising that medical academies and physicians were the main prime movers of early meteorological observations in Spain.

Navarrete was author of many works, most of them unedited and kept as manuscripts in the archives of the Spanish academies of Medicine and History. His attention was focused not only on medicine, but also on physical observations, cosmography, geography, botany, and, in general, all the fields included in the 'natural history' discipline. His main work was *Ephemérides barométrico-médicas matritenses* ('Ephemeris barometric-medical from Madrid'), published in Madrid in 1737 (this text is digitized and available at the Library of the Seville University, http://fondosdigitales.us.es). It is a set of daily meteorological
observations (atmospheric pressure, temperature, wind direction, qualitative comments on rain, cloudiness, and other meteorological events) taken in Madrid from March to November 1737. Here, the author establishes the basis of an observational program dedicated to compile all the meteorological data potentially useful to medical studies, not only in Madrid, but also in other Spanish cities. Unfortunately, this program was not accomplished due to the lack of interest of the authorities (Anduaga Egaña, 2012).

A precedent of the *Ephemerides* is the book studied in this paper, *Cielo y suelo granadino* (*Sky and soil in Granada*). The manuscript is dated on 1732, and, although finally it was not published, the book was finished and prepared for publication. It is kept in the Library and Archive of the Franciscan Province of Cataluña, Barcelona, and it has been edited recently (Gil Albarracín, 1997). Among the multiple aspects of natural history studied by the author, we are interested on the climatological and meteorological observations. The chapter IV is entitled 'Observations of the atmosphere using the barometer and the
thermometer', and includes monthly summaries (with daily resolution) of these observations from December 1728 to February 1730. The chapter XVI is entitled 'Medical observations for the knowledge of climate'. Here, the author offers, at monthly and/or seasonal resolution, a summary of climatic conditions from 1706 to 1730 in Granada, as well as their relationships with the appearance of illness in the city, following the neo-Hippocratic paradigm. We establish the beginning of the qualitative series in 1706 because the author, in the description of the cold winter 1729, indicates that this year was the "coldest winter
seen in twenty four years", suggesting that he began to compile his observations that year. These data are available at the data repository of the University of Almería (Rodrigo, 2018; file 'NavarreteData.xlsx', http://hdl.handle.net/10835/6248). In next sections we study separately both chapters, because the time resolution, and the nature of data, are different in each case.




### 3 The period 1706-1730.

The chapter XVI of the book by Navarrete (pages 105-107 of the manuscript) is dedicated to expose the 'alterations of health due to mutations of the air, and general causes obtained from the long observation and practical knowledge of the country'. Here, the author establishes relationships between different diseases and climatic conditions. So, for instance, in the thirteen
paragraph (page 106v), he says that "If cold, rain and snowfalls continue until May: difficult births, chest pains, and dangerous anginas: year 1727". This paragraph allows to characterize the spring of 1727 as wet and cold. The analysis of the contents of this chapter yields as result the summary shown in Table 1, where the seasons unmistakably cold, warm, wet and dry are indicated (in the following we designate these seasons as 'extreme seasons'). Seasons are defined as usual: winter (December, January, February, winters identified by the year corresponding to January and February), spring (March, April, May), summer
(June, July, August), and autumn (September, October, November).

In a first view, it seems that there is a predominance of cold over warm conditions in winter and spring, and dry over wet conditions in all the seasons, except spring. Some of these extreme seasons are confirmed by other data sources. So, for instance, cold winters 1709, 1729, and 1730 have been reported in other Spanish cities, as Tortosa, Seville, and Alicante (Alberola Romá, 2014), as well as the drought during the 1720s decade, in Jerez de la Frontera (AHVM, 1722), Arcos de la
Frontera (ACAF, 1723), and Sevilla (Zúñiga, 1747), where *pro-pluvia* rogations were celebrated.

Documentary data normally provide information on extreme events. In a first step, it is possible to obtain a catalogue of episodes like droughts, intense rainfalls, snowfalls, hailstorms, etc. A preliminary view of this catalogue may be misleading, the risk is to consider that these events were the 'normal' climatic conditions in the studied period. The usual methodology, based on ordinal indices (Brázdil et al., 2010), maintains this view if there is not an appropriate overlapping period between
documentary and instrumental data to calibrate and validate the index  (this is our case, given the brief period analysed). The question is if the historical frequency of extreme seasons is exceptional, or, in the opposite, may be considered as 'normal' according to 20[th] century standards.

Rodrigo (2008) proposed a methodology alternative to indices, trying to overcome the problem of the lack of an overlapping period. This method was tested using climate model paleo simulations (Rodrigo et al., 2012). If $p_{10}$ and $p_{90}$ are the percentiles
10 and 90 of a climatic series X of mean value u and standard deviation s, we can find corresponding values normalized $q_{10}$ and $q_{90}$,

$$q_i = \frac{p_i - u}{s} \qquad i=10, 90 \tag{1}$$

The percentiles $q_i$ (i=10, 90) correspond to the standard normal distribution $F_X$. The normality hypothesis is the simplest choice, and it is valid for the series of temperature and rainfall in the four seasons of the year, except in the case of summer rainfalls
(Rodrigo et al., 2012). We can obtain the values $q_i$ from the number of extreme seasons $n_i$, with n=25 (number of years of our series), that is,

$$\frac{n_{10}}{n} = \text{Prob}\{X \le q_{10}\} \to q_{10} = F_X^{-1}\left(\frac{n_{10}}{n}\right) \tag{2}$$

$$\frac{n_{90}}{n} = \text{Prob}\{X > q_{90}\} = 1 - \text{Prob}\{X \le q_{90}\} = 1 - F_X(q_{90}) \to q_{90} = F_X^{-1}\left(1 - \frac{n_{90}}{n}\right)$$

From equation (1), we can express the corresponding standard deviation s, and mean value u as

$$s = \frac{p_{90} - p_{10}}{q_{90} - q_{10}} \qquad u = p_{10} - sq_{10} = p_{90} - sq_{90} \tag{3}$$



The basic idea is to accept that threshold values $p_i$ (obtained from the instrumental observations) are also valid to define extreme values in past, that is, we accept that during a past extreme season the value of the climate variable X was lower (higher) than $p_{10}$ ($p_{90}$). Percentiles 10 and 90 are commonly used to define the frequency of extreme indices, such as cold nights or warm days, and correspond to moderately extreme events (Zhang et al., 2005). Summarizing, from documentary data analysis, the

numbers of extreme seasons $n_i$ (i=10, 90) are obtained (Table 1). These numbers are used to estimate $q_i$ (equation 2), and the s and u values are calculated considering the values $p_i$ corresponding to the instrumental period (equation 3). The hypothesis here is that climatic changes are revealed not only by changes in the mean value of the variables, but also in the frequency and intensity of extreme events. Therefore, if we know the frequency of extremes during a given period, and accepting the normality hypothesis, we can determine the mean value and standard deviation of the climate variable corresponding to that

period.

The reconstruction of s and u depends of the values $p_i$ previously established as threshold values to define extreme seasons. These values may be established using the percentiles 10 and 90 corresponding to a given reference period. Therefore, the reconstruction is strongly dependent on the chosen reference period. A possible solution is to select as reference period a period in which there are different climatic situations. Here we use the period 1895-2005, which contains years characterized by a

weak warming signal (first decades), and years with a clear warming signal (last decades of the 20[th] century).   Temperature data are extracted from the database Spanish Daily Adjusted Temperature Series (SDATS, Brunet et al., 2006). Monthly rainfall data are extracted from the database made by the Spanish Agency of Meteorology (AEMET, Luna et al., 2012). These databases are available at the web page of the AEMET (http://www.aemet.es). All the series are homogeneous and do not present missing data or gaps. Table 2 shows the percentiles 10 and 90 of seasonal mean temperature and accumulated

precipitation in Granada corresponding to the complete period 1895-2005.

To calibrate the method, the complete series was divided into 25-year running periods, the first one being 1895-1919, the second one 1896-1920, until the last period 1981-2005.  This procedure was adopted to obtain a large empirical sample. For each individual period, the mean value u and the standard deviation s were estimated applying the method, that is, from the number of extreme seasons $n_{10}$ and $n_{90}$. Results were compared with the corresponding u and s observed values: correlation

coefficients between estimated and observed values, as well as the root-mean-squared error (RMSE), were calculated. RMSE is used in forecasts verification and can also be thought as a typical magnitude for forecast errors (Wilks, 1995). Values of RMSE were used to provide an estimate of the uncertainties that are associated with the reconstruction methodology. Table 3 shows the results of this calibration. All the correlation coefficients were significant at the 95% confidence level. According to correlation coefficients values, the method offer better results for mean value u (standard deviation s) of temperature

(rainfall). These differences may be due to deviations from normality in the case of rainfalls, particularly in summer. Figure 2 shows as example the comparison for the autumn rainfall series.

The method was applied to the historical period 1706-1730, using data of Table 1 as $n_i$, and the percentiles $p_i$ of the reference period (Table 2). Figures 3 and 4 and Table 4 show the reconstruction of seasonal temperature and accumulated rainfall distribution functions, accepting the normality hypothesis. Only in the case of summer rainfall the reconstruction was not

accomplished, because of the absence of extreme wet seasons (Table 1), and the non-normal character of summer rainfalls. RMSE values previously estimated are used as error margins. Results are compared with the corresponding values of two 25-year periods in the 20[th] century, 1906-1930, and 1976-2000, when the global warming signal is very different. To obtain a best view of this comparison, Table 5 shows the statistics corresponding to these periods. According to these results, seasonal mean temperatures during 1706-1730 were very similar to those during 1906-1930, even slightly warmer, and lesser (except in

summer) than temperatures during 1976-2000, around 0.7 °C in winter, 0.4 °C in spring, and 1 °C in autumn. Standard deviations of temperature during 1706-1730 was similar to 1906-1930, and lesser than that of 1976-2000, suggesting smaller variability in the past. Total rainfall shows values very similar in autumn for the three periods, slightly wetter conditions in





spring during 1706-1730 and 1906-1930, and slightly wetter conditions in winter of 1706-1730 in comparison with 1906-1930. Variability of rainfall in 1706-1730 is similar to that in 1976-2000, except in spring, characterized during 1976-2000 by drier conditions.

These results suggest that during 1706-1730 temperatures were very similar to those of the first decades of the 20th century, when the warming signal may be considered very small in comparison with the last decades of the 20th century. In relation to rainfall, there are not clear differences between periods, except in spring of 1976-2000, when there were drier conditions than in the past. We must note that the period 1706-1730 is immediately subsequent to the cold Maunder Minimum. Luterbacher *et al*. (2004, 2007) and Xoplaki *et al*. (2005) found a warming trend in European Winter and Spring temperatures from the late Maunder Minimum, culminating in the late 1730s (the mean value of the autumn temperature in Central England between 1729 and 1738 was 10.5 °C, equal to that recorded during 1991-2000, Jones and Briffa, 2006). Warming from the markedly cold decade of the 1690s to the 1730s is probably due to the scarcity of major explosive volcanic eruptions from the early 1700s compared to the previous two decades (Jones and Briffa, 2006).

According to dendroclimatological studies (Manrique and Fernández-Cancio, 2000), the main phase of the Little Ice Age in Spain corresponds to the 16th and 17th centuries, and the 18th century marks the beginning of a recovering of conditions towards 20th century standards. This result also has been found by Spanish climate historians (Font Tullot, 1988; Alberola Romá, 2014). Therefore, the Little Ice Age was not a continuous and homogeneous cold period in southern Spain, but it was characterized by the alternation of different phases, reflecting a high climate variability. This high variability also has been recorded from dendroclimatological studies covering the whole Mediterranean Basin (Nicault et al., 2008).

**4 From December 1728 to February 1730.**

The chapter IV of the book (pages 12-16 of the manuscript) is entitled 'Observations of the atmosphere using the barometer and the thermometer'. It is the first compilation of early instrumental meteorological data in Spain, so far as we know. It begins in December 1728, and ends in February 1730. The author explains that he shows his observations of 1729 as an example of the effects of atmospheric variability, and that these observations 'are broadly in line with the observations that I have taken during nine years with these instruments to determine the conditions of the atmosphere'. Unfortunately, we have not found documentary sources with these nine years of data, and we have to be content with the available information. In addition, information is presented as monthly summaries, indicating characteristic values or corresponding to critical moments, and not cover in detail all the days of the period. Sometimes, he adds comments on winds and other meteorological events (fog, cloudiness), and he indicates the number of rainy days of some months. The instruments used by Navarrete were an 'English barometer' and a 'Florentine thermometer'. There is no information about the installation of the instruments or the exact time at which readings were taken, and in the case of temperature, the scale does not correspond to any of the better-known scales that were introduced later (for instance, the Reamur scale). This means that any values measured are only important in relative terms (Brázdil et al., 2008). Nevertheless, we have tried to 'calibrate' these observations using the information provided by the observer.

Navarrete used a Florentine thermometer with 'spirit of wine' as thermometric liquid. After a brief description of the instrument, he explains how established the scale used to measure temperatures: he distinguishes between 'maximum cold', in the extreme cold of winter or when the 'little bottle was buried in snow with salt ammoniac', and 'maximum heat', in the extreme warm summer, or 'in the mouth of an oven'. Navarrete marks 'maximum cold' with the value T=100, and 'maximum heat' with the value T=1, and divides the length of the thermometer in equal divisions, calling 'equilibrium' to the intermediate value T=50. The lower defining point of the Fahrenheit scale (0 °F = -17.78 °C) was established as the temperature of a solution of brine made from equal parts of ice, water and ammonium chloride (Fahrenheit, 1724). Note that the 'maximum cold' was established by Navarrete in a similar way, although, unfortunately, he does not indicate the proportion of salt nor the alcohol



content of the thermometric liquid. In the chapter V ('What can be deducted from these observations') Navarrete explains that these limits correspond to 'regular conditions', but they may be exceeded. Figure 5 shows the measures recorded by Navarrete from December 1728 to February 1730. The sensitivity or resolution of the scale is 0.5 degrees (on 12 July 1729 Navarrete recorded T = 38 ½ degrees, and from 26 to 28 December 1729, T = 87 ½ degrees). The author indicates the appearance of

frosts on 25 December 1728 (T = 90), 28 December 1728 (T = 99) and 19 February 1729 (T = 98), and explains that on 2 February 1729, when the thermometer indicated T = 86, 'ice melted'. We estimate the minimum value indicated T = 90 as the threshold value of the occurrence of frosts. In relation to the 'equilibrium' (T = 50), Navarrete indicates that 'it is normal that during the month of May cold and heat equalize, on 29 May the thermometer reached the exact average value'.

We do not know the exposure conditions and the time of the day in which measurements were taken. However, some

information may be obtained from the analysis of the text. In particular, when the author describes the month of July, he explains that 'the first day, the thermometer exposed to the sun at the nap hour increased from 39 to 12'. Given the magnitude of other measurements (for instance, T = 34 on 25 July 1729, 'the warmest day of the year'), we can infer that measurements were taken sheltered from the solar radiation (probably indoor), in the afternoon ('nap hour'). Therefore, these measurements may be considered as proxy of daily maximum temperatures (Camuffo, 2002; Wheeler, 1995).

We have tried to calibrate these measurements accepting a linear relationship between the scale used by Navarrete and the Celsius scale (Vittori and Mestitz, 1981). For calibration, taking into account the previous comments, we assign $0.0 \pm 0.1$ °C to T = 90.0 $\pm$ 0.5 (frosts), and $23.3 \pm 0.1$°C (mean value of daily maximum temperature corresponding to May during the reference period 1906-1930, and standard error at the 95% confidence level) to T = $50.0 \pm 0.5$ ('equilibrium'). This last hypothesis is based on results of the previous section that indicated the similarity between temperatures of the historical period

1706-1730 and 1906-1930. The calibration equation is

$$°C = aT+b \qquad\qquad\qquad (4)$$

Using the law of propagation of uncertainty, parameters of the equation are a = $-0.58 \pm 0.02$ °C/T, and b = $52 \pm 2$ °C. Applying this equation we obtain that T = $100.0 \pm 0.5$ ('maximum cold' recorded) is equivalent to $-6 \pm 4$ °C, and T = $34.0 \pm 0.5$ ('maximum heat' recorded) to $32 \pm 3$ °C. The value T = $12.0 \pm 0.5$ (recorded on 1 July 1729 at the afternoon and with the

thermometer exposed to solar radiation) would be equivalent to $45 \pm 2$ °C. These values are plausible: 'maximum cold' (obtained when thermometer is in a bath of snow with salt ammoniac) must correspond to a temperature below 0 °C (due to the freezing-point depression of a salt solution), the mean value of daily maximum temperatures in July is $32.7 \pm 0.1$ °C, and the absolute daily maximum temperature is $40.9 \pm 0.1$ °C during the reference period 1906-1930.

The equation (4) was applied to the daily temperatures recorded by Navarrete, and afterwards the monthly mean values were

estimated, and compared with the monthly mean value of daily maximum (mean) temperatures TX (TM) recorded during the period 1906-1930. Results are showed in Fig. 6. It may be seen that conditions were colder than modern reference values in winter 1729, autumn 1729, and winter 1730, even with values lower than reference period TM values. From May to August, however, reconstructed values and their margin errors, match with modern TX values. Roughly speaking, these results agree with the qualitative comments made by Navarrete in the chapter XVI, where he describes winter 1729, autumn 1729, and

winter 1730 as cold seasons, spring 1730 as warm season, and he does not indicate particular conditions for summer 1730, which, in consequence, it may be considered as a 'normal' season.

It deserves special attention the winter 1729, 'the coldest winter seen in 24 years' according our author. Figure 7 summarizes quantitative and qualitative observations taken during this winter: first days of December 1728 was dominated by a 'cold fog' and high pressures. A sharp decrease of pressure marked the snowfall on 13 December, and three consecutive rainy days from

19 to 21 December. Frosts, rainfall, snowfalls, hail, and northern winds characterized the last days of this month, with T =





100.0 ± 0.5 (-6 ± 4 °C, according our calibration) on 29 December. After cloudy days on 30 and 31 December, four snowy days (on 7, 12, 13, and 18 January) were recorded (in the reference period the mean value of snowy days is 0). Ice and snow stayed 'in shadow places' until 2 February, when it rained. During February 'fog, sun and frosts continued'. Temperatures indicated by the author during this winter were colder than T = 78.0 ± 0.5 (7 ± 4 °C). Figure 8 shows the monthly average sea level pressure field (SLP, left), and anomalies of the SLP field with respect to the reference period (right) according to the independent reconstruction by Luterbacher et al (2002), available at http://climexp.knmi.nl. Anticyclonic conditions, especially during February, made possible the appearance of frosts and morning fogs, with northwestern winds. The negative anomalies corresponding to December and January would explain the predominance of rainfalls and snowfalls between mid-December and mid-January.

Atmospheric pressure was measured using an English barometer. The observer was more interested on the fluctuations of this variable than on absolute values. So, sometimes, he records deviations with respect to a mean value, which it is not specified (in the 20th century reference period, the annual mean value of pressure in Granada is 939 hPa, of order of 28 English inches). A deviation of 1 line means changes of order of 3 hPa. Barometers usually had a mobile scale with qualitative marks (Guijarro, 2005), from the highest value ('Very Dry') to the lowest value ('Very wet'). The number of quantitative measurements is scarce, and we do not know the exposure conditions nor the temperature of the barometer, in consequence it is impossible to apply the usual correction to 0°C. Information on atmospheric pressure is basically qualitative, with references to 'Very dry', 'Good Weather', 'Variable', 'Windy and/or Wet', and 'Very Wet' categories. 'Very dry' conditions are recorded on 12 December 1728, with a positive deviation of 4 lines above the mean line, that is, around 12 hPa. On 13 December, according to the author, 'the thermometer and the barometer fell down four lines in the morning, I predicted snow, it arrived soon, it was a lot of snow, and persisted all the day'. The categories 'Variable' and 'Good Weather' are associated to pressure values 1 line above the mean value (for instance on 25 April 1729, and 17 January 1730). The class 'Windy and/or Wet' indicated by the barometer is associated to information on snowfalls (27 December 1728), strong rainfalls (26 September 1729), or intense rainfalls accompanied by westerly winds (30 November 1729). On 29 December 1728 the barometer indicated 'Very Wet' conditions ('it rained a lot, and hailed'). Therefore, pressure information is related to other variables (snowfalls, rainfalls, winds). Sometimes, the author summarizes the general behavior of a concrete month, for instance when he indicates that during April 1729 'westerly winds continued, with clouds and water, well-marked by the barometer'. This month it rained on days 1, 2, 8, 11, 13, 14, and 23, seven rainy days, coinciding with the average value of days with rain higher than 1 mm during the reference period 1971-2000 (INM, 2004). Note that rainfall information is often accompanied by information on west winds, and cold weather is associated to north winds. South winds are associated to hot conditions (for instance, on 29 May 'flew a southeast wind and the afternoon was hot', and the author indicates southwest wind on 25 July 1729, 'the warmest day of the year'). As we know, from the analysis of the 20[th] century climate in the Iberian Peninsula, westerly flow in winter is connected with a higher percentage of extreme precipitation, and cold extremes are associated to the advection of cold air masses from the north (Fernández-Montes et al., 2012). On the other hand, a great part of warm days in spring and summer is related to southern flows (Fernández-Montes et al., 2013). Therefore, although the information yielded by Navarrete is scarce, it seems coherent with climatic observations based on instrumental data in the 20[th] century.

**5 Conclusions.**

In this work we have reconstructed the climatic mean conditions during the first third of the 18[th] century in Granada (southern Spain) using documentary data. In addition, we have retrieved a new early meteorological data series, probably the first instrumental series measured in Spain. As result of this work, some conclusions can be obtained:

- Seasonal temperature and rainfall during the historical period 1706-1730 were very similar to those corresponding to the 1906-1930 period, at the beginning of the 20[th] century, when the global warming signal was of less importance.





The first decades of the 18[th] century can be characterized as a period of certain recovery after the cold Maunder Minimum period.

- Some important extreme events were detected, as the drought in the first half of the 1720s decade, and the cold wave during the winter 1729.
- The original temperature scale was calibrated and converted to the Celsius scale, obtaining plausible values, which, at daily and monthly time scale, allow characterize the annual cycle of temperature in 1729.
- The reconstruction is coherent with independent reconstructions of past climates, in particular, the sea level pressure field in Western Europe.

More research is needed to complete our view on past climate conditions. In particular, it is hoped that more daily instrumental
observations and weather registers may eventually come to light. The enlargement of databases, and the study of documentary data and early instrumental data, may contribute to the knowledge of natural climate variability and, therefore, to the understanding of climate processes.

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





Zúñiga, B.L.: Anales eclesiásticos y seglares de la M.N. y M.L. ciudad de Sevilla: que comprenden la Olimpiada ó Lustro de la Corte en ella; con dos Apéndices, uno desde el año de 1671 hasta el de 1728, y otro desde 1734 hasta el de 1746. Biblioteca de Andalucía, Sgn.: ANT-XVIII-470, 1747.





Table 1. Extreme seasons in Granada from 1706 to 1730.

| Season | Cold | Warm | Wet | Dry |
|---|---|---|---|---|
| Winter | 1709 1723 1729 1730 | 1719 | 1718 1725 1729 1730 | 1719 1720 1721 1722 1723 1724 |
| Spring | 1726 1727 | 1729 | 1707 1719 1721 1725 1727 | 1718 1722 1724 |
| Summer | 1710 1728 | 1719 1726 | | 1707 1718 1719 1720 1722 1724 1726 |
| Autumn | 1729 | 1726 | 1725 1728 1729 | 1707 1718 1722 1724 |

Table 2. Percetiles 10 and 90 of seasonal mean temperature and total rainfall in Granada from 1895 to 2005.

| | Temperature (°C) | | Rainfall (mm) | |
|---|---|---|---|---|
| | $p_{10}$ | $p_{90}$ | $p_{10}$ | $p_{90}$ |
| Winter | 6.1 | 8.6 | 57.7 | 218.0 |
| Spring | 12.0 | 14.7 | 52.3 | 186.3 |
| Summer | 22.7 | 25.1 | 2.2 | 49.6 |
| Autumn | 14.5 | 17.1 | 48.0 | 161.0 |

5   Table 3. Calibration of the reconstruction methodology using 25-year moving series from 1895 to 2005. u = mean value; s = standard deviation; RMSE = root-mean-square error; r = correlation coefficient between observed and estimated parameters.

| | Temperature | | Rainfall | |
|---|---|---|---|---|
| | RMSE (°C) | r | RMSE (mm) | r |
| u(winter) | 0.1 | 0.96 | 17 | 0.47 |
| s(winter) | 0.08 | 0.66 | 6 | 0.93 |
| u(spring) | 0.2 | 0.90 | 6 | 0.79 |
| s(spring) | 0.07 | 0.56 | 5 | 0.80 |
| u(summer) | 0.2 | 0.80 | 5 | 0.49 |
| s(summer) | 0.2 | 0.92 | 14 | 0.76 |
| u(autumn) | 0.2 | 0.94 | 6 | 0.51 |
| s(autumn) | 0.5 | 0.76 | 3 | 0.92 |





Table 4. Reconstruction of the period 1706-1730 in Granada. u = mean value; s = standard deviation.

|  | Temperature | | Rainfall | |
|---|---|---|---|---|
|  | u(°C) | s(°C) | u(mm) | s(mm) |
| Winter | 7.0 ± 0.1 | 0.91 ± 0.08 | 124 ± 17 | 94 ± 6 |
| Spring | 13.2 ± 0.2 | 0.86 ± 0.07 | 130 ± 6 | 66 ± 4 |
| Summer | 24.0 ± 0.2 | 0.8 ± 0.2 |  |  |
| Autumn | 15. 8± 0.2 | 0.7 ± 0.5 | 100 ± 6 | 52 ± 3 |

Table 5. Statistics of the periods 1906-1930 and 1976-2000 in Granada. u = mean value; $I_u$ = 95% confidence level interval for mean value; s = standard deviation; $I_s$ = 95% confidence level interval for standard deviation.

|  | Temperature | | Rainfall | |
|---|---|---|---|---|
|  | 1906-1930 | | | |
|  | u ($I_u$)  (°C) | s ($I_s$) (°C) | u ($I_u$)  (mm) | s ($I_s$)  (mm) |
| Winter | 6.9 (6.3;7.2) | 0.8 (0.7;1.1) | 106 (92;120) | 33 (26;43) |
| Spring | 12.8 (12.5;13.1) | 0.8 (0.7;1.1) | 130 (112;148) | 42 (35;56) |
| Summer | 23.4 (23.1;23.7) | 0.8 (0.7;1.1) | 23 (15;31) | 19 (15;25) |
| Autumn | 15.3 (14.9;15.7) | 0.9 (0.7;1.2) | 113 (96;130) | 41 (33;54) |
|  | 1976-2000 | | | |
|  | u ($I_u$)  (°C) | s ($I_s$) (°C) | u ($I_u$)  (mm) | s ($I_s$)  (mm) |
| Winter | 7.7 (7.3;8.1) | 0.9 (0.7;1.2) | 129 (90;168) | 95 (74;132) |
| Spring | 13.6 (13.2;14.0) | 1.0 (0.8;1.4) | 97 (77;116) | 47 (37;65) |
| Summer | 23.9 (23.3;24.5) | 1.5 (1.2;2.1) | 20 (12;29) | 20 (16;28) |
| Autumn | 16.8 (15.6;17.2) | 2.0 (1.6;2.8) | 108 (90;125) | 42 (33;58) |





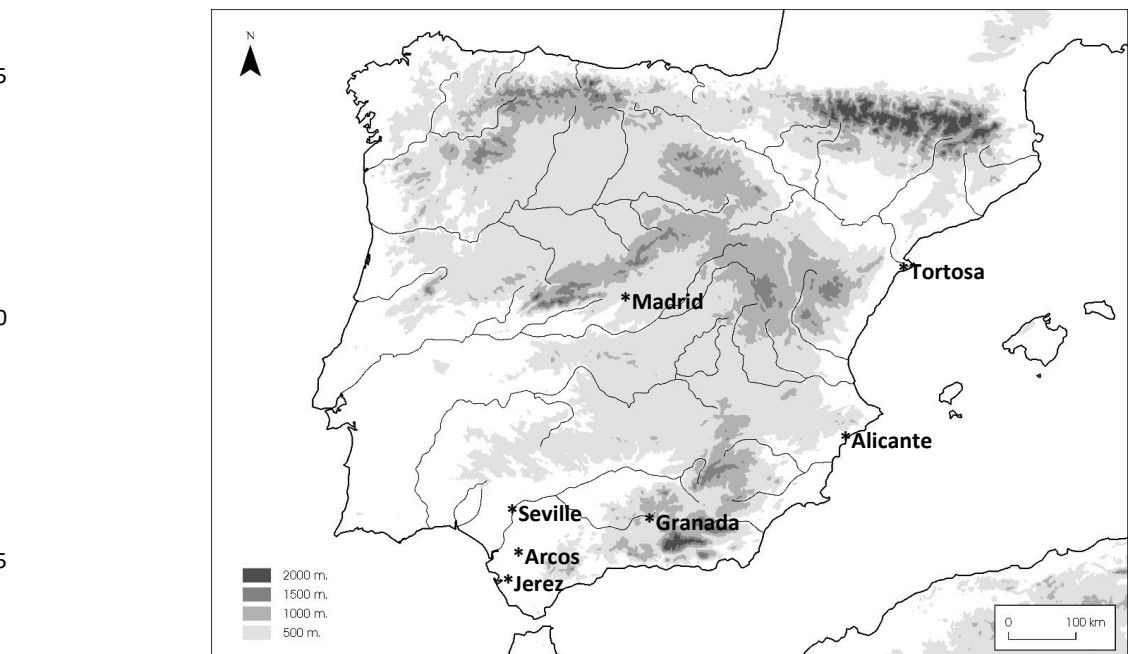

Figure 1. Location of Granada and other cities mentioned in the text.





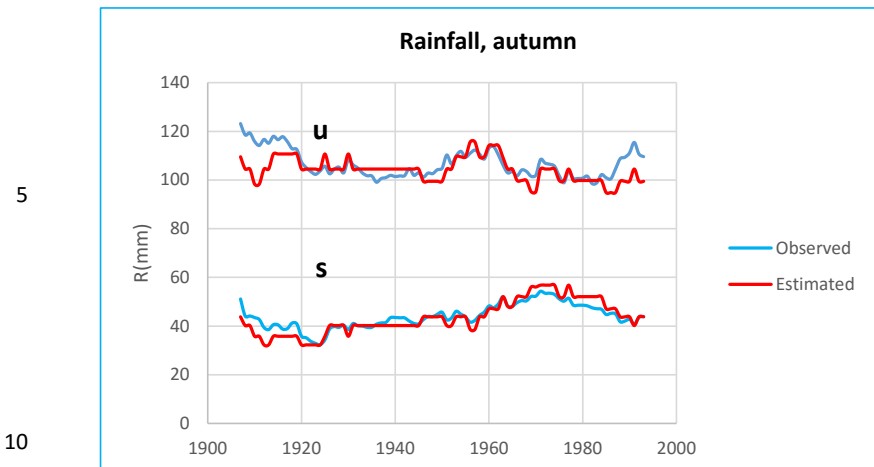

Figure 2. Calibration of the reconstruction method for autumn rainfall. u = mean value; s = standard deviation.





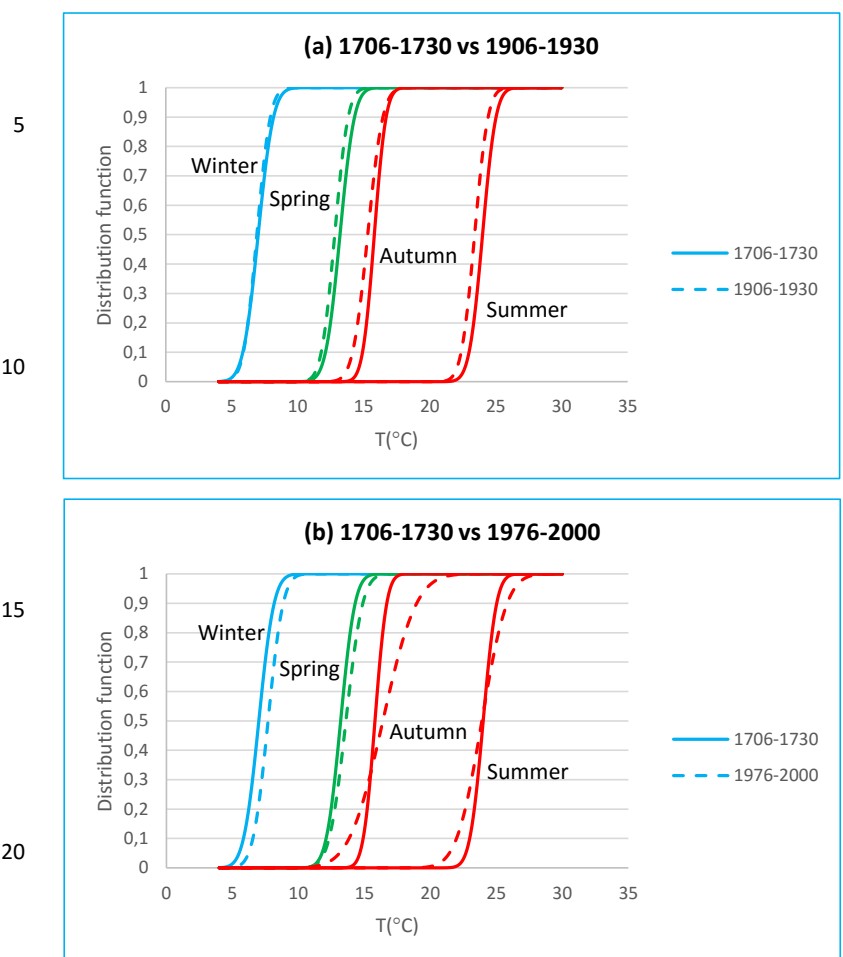

**Figure 3. Distribution functions of seasonal temperatures of 1706-1730 and comparison with 1906-1930 (a), and 1976-2000 (b).**





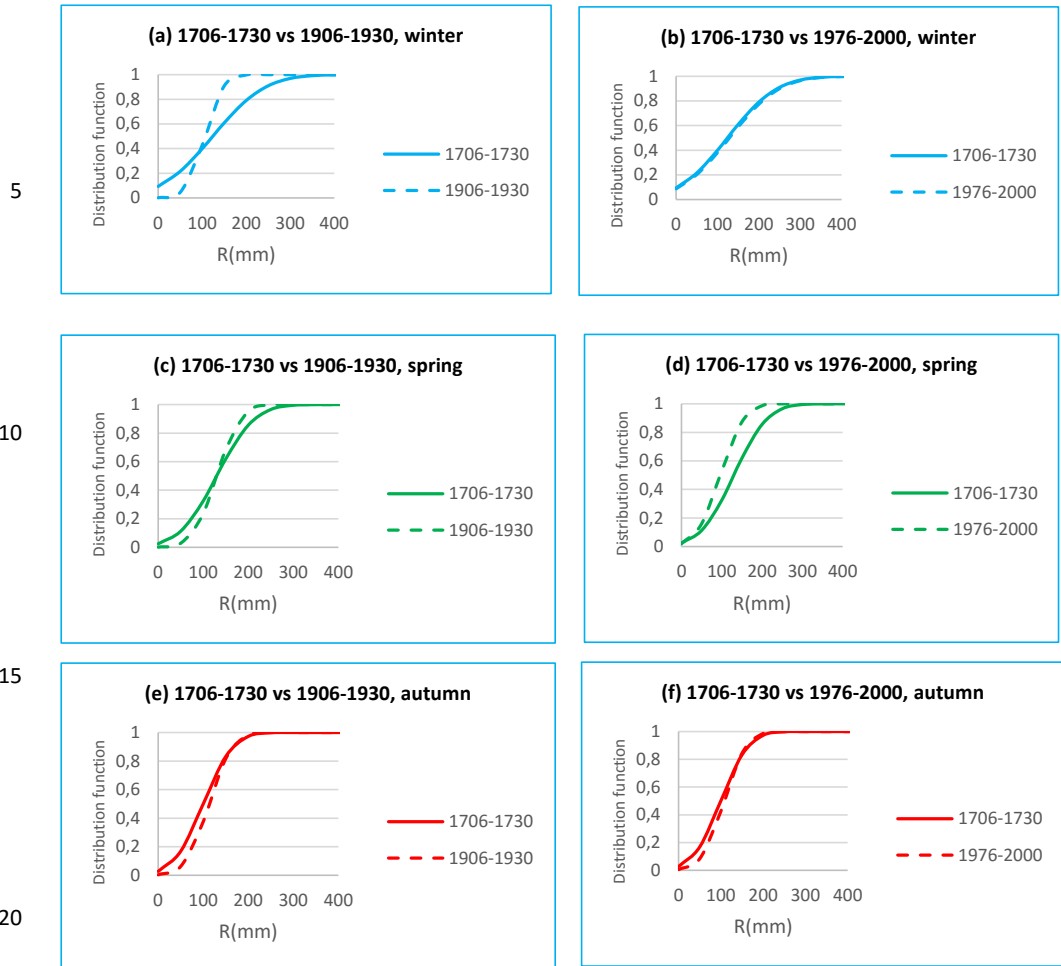

**Figure 4. Distribution functions of seasonal rainfall of 1706-1730, and comparison with 1906-1930 (a, c, e), and 1976-2000 (b, d, f).**

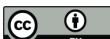



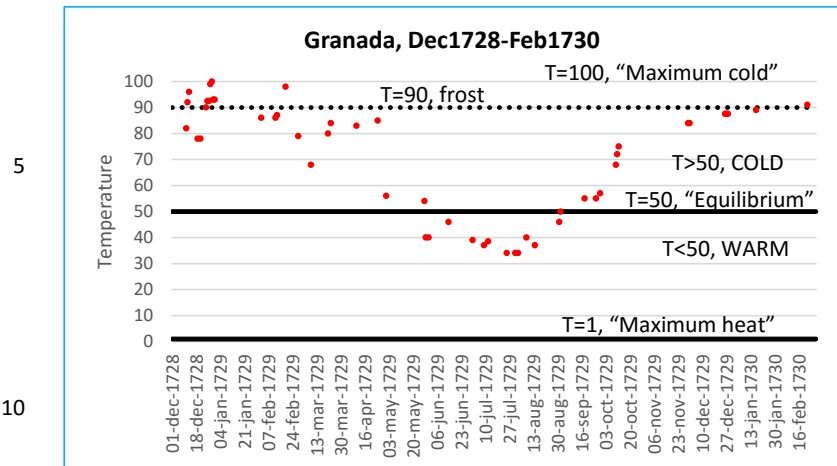

**Figure 5. Temperatures measured and scale defined by Navarrete with the Florentine thermometer.**





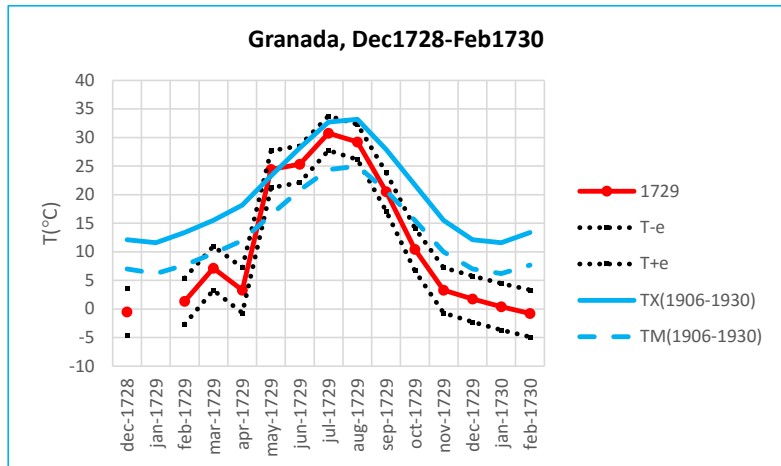

**Figure 6. Monthly mean value of daily temperature in 1729 and error margins estimated, and comparison with monthly mean value of daily maximum temperature (TX) and monthly mean value of daily mean temperature (TM) of 1906-1930.**



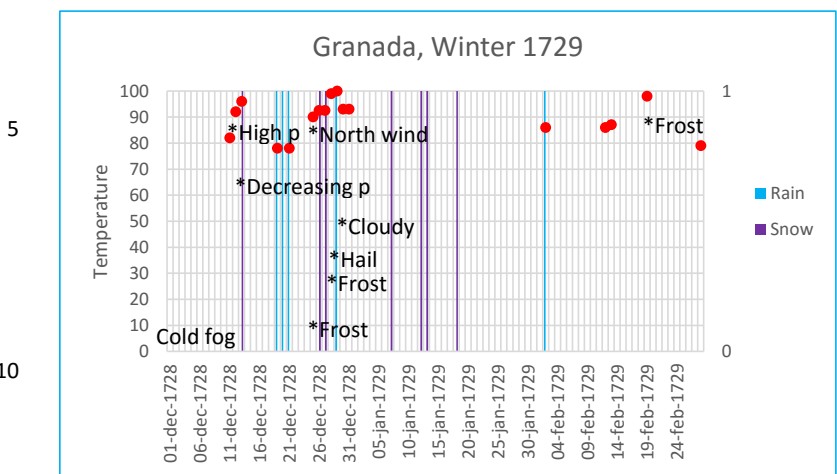

**Figure 7. Observations during the winter 1729. Left axis (dots): temperature according to Navarrete's scale. Right axis: rainy and snowy days.**





**Figure 8. Reconstruction of the monthly SLP field in Western Europe (left) and anomalies of the monthly SLP field with respect to the reference period 1906-1930 (right) for December (a, b), January (c, d), and February (e, f), according to Luterbacher et al (2002).**