# Peer review of "The climate of Granada (southern Spain) during the first third of the 18th century (1706-1730) according to documentary sources."

_Climate of the Past, 2018_

## Referee Comment (RC1) · Anonymous Referee #1 · 8 Feb 2019

General comments

The discussion paper titled "The climate of Granada (southern Spain) during the first third of the 18th century (1706-1730) according to documentary sources" is very stimulating and makes a valuable contribution to historical climatology. This article is undoubtedly within the scientific field of CP.

The major contribution and novelty of this paper is the reconstruction of the climatic mean conditions during the first third of the 18th century in southern Spain based on new documentary data and an early and original meteorological data series. In addition, the author uses a relatively recent statistical method to exploit data.

[Figure]

Title and summary reflects the content of the article. Many references to previous researches are used and relevant. Contributions of various authors are clearly outlined. The overall presentation is clear and well structured with a fluent and precise language (for a non-native english reader). Substantial conclusions are reached and based on a reproductible methodology and calculations.

Specific comments

As a historian, my comments are principally focused on this matter and may not be relevant on other subjects:

1) The summary reflects the content of the article but should perhaps more clearly highlight its main contribution and novelty, i.e. applying a methodology alternative to Pfister-indices to a new and original set of documentary data.

2) Documentary data are available for a period inferior to three decades. As climatological normals are used as baseline to evaluate climate events and provide context for year-to-year variability, is it a weakness for analysis and statistical comparisons?

3) Before using the methodology alternative to Pfister-indices, would it not be useful to establish the robustness of Navarrete's observations by comparing the indices drawn from his work with other available series?

4) In the same logic, after application of the method proposed by Rodrigo (2008), would a cross comparison between the reconstruction made and another series (1906-1930 and 1976-2000) not be useful to strenghten the evidence?

5) Is the observation program established by Navarrete in 1737 original and how does it fit into the cultural context of the time?

Technical corrections

- Page 2, line 25: "Precursor" or "Archetype" rather "A precedent"?

- Page 4, line 28: "All the correlation coefficients were significant at the 95% confidence

level." A statement to reformulate?

- Page 8, line 1: "a period of certain recovery", is the medical metaphor relevant? Perhaps "transition to a new phase after the cold Maunder Minimum period."?

- Page 8, lines 9-10: A stronger conclusion would be useful to highlight the contribution of the article on a poorly documented period for Spain?

- Table 1: Addition of a temporal comparison with another space or very precises rogations series available (https://www.clim-past-discuss.net/cp-2018-67/ for exemple) is perhaps relevant?
* * *

---

## Author Comment (AC1) · 11 Feb 2019

Reply to Referee#1

Thank you very much for your comments and suggestions. They will be taken into account in the revised version of the manuscript. In relation to your specific comment 2, I think that a brief comment is necessary. The proposed methodology does not try to provide the year-to year variability, but the general characteristics of the studied period. In fact, the paper by Rodrigo (2008) is entitled "A new method to reconstruct low frequency climatic variability from documentary sources: application to winter rainfall series in Andalusia (southern Spain) from 1501 to 2000". If we apply the method to a

longer time period, using 25 or 31 years moving periods, we obtain the running means of the series, not values year-to-year. However, if instrumental series begins at the end of documentary series (for instance, if documentary data end at 1800, and instrumental data begin at 1801), it is possible to obtain these annual values. The procedure is explained in Rodrigo et al (2012). Past annual values $x(t-r)$ may be obtained by means of a backward iterative process starting with instrumental values $x(t+r+1)$ and using running means ($u(t)$ for the documentary period, $u(t+1)$ for the instrumental period), with $n=2r+1$,

$$x(t-r) = x(t+r+1) - n(u(t+1) - u(t))$$

Unfortunately, continuous instrumental series in Granada do not appear until mid-19th century, and, in consequence, it is not possible to apply here this extension of the method.

It is true that this is a weakness of the analysis. However, this methodology has advantages in comparison with the standard indices methodology. First, ordinal indices may be skewed by the subjectivity of the authors in original sources, and/or by the interpretation of the researcher of the descriptions in the sources. In addition, ordinal indices are often based on the impact of the events on the socioeconomic infrastructures (for example, destruction of bridges during a river flood, loss of harvests, etc), and these impacts may change in different historical periods. The risk here is to consider as heavy extremes certain events that show the vulnerability of the system more than the extreme character of climate variables. The method followed is not based on the severity of the phenomena, and, in consequence, at least to a certain degree, avoid these problems. In second place, does not need an overlapping period with instrumental data, which is necessary to calibrate and validate indices. There is a third problem of statistical character: the calibration of indices normally is made using a regression procedure between proxy data (indices) and instrumental data during the overlapping period. From a statistical point of view, the consequence is the loss of variance of the reconstructed series, a problem that normally is solved using an 'inflation factor' to

correct the reconstructed series (this is a typical problem in the management of proxy data, you can see how it was applied to documentary data in Rodrigo and Barriendos (2008)). With this method, in principle, it is not necessary to introduce this mathematical artefact (see Tables 4 and 5 of the manuscript, particularly the columns 's' showing the standard deviation of the reconstructed series and instrumental series). Statistical comparisons are made using the 95% confidence level intervals for mean value and standard deviation (Iu and Is in Table 5, respectively).

These explanations were not included in the manuscript to avoid an excessively detailed treatment of the methodology, which is explained in the references quoted:

Rodrigo, F.S.: A new method to reconstruct low-frequency climatic variability from documentary sources: application to winter rainfall series in Andalusia (southern Spain) from 1501 to 2000, Clim. Change, 87, 471-487, 2008.

Rodrigo, F.S., Gómez-Navarro, J.J., and Montávez-Gómez, J.P.: Climate variability in Andalusia (southern Spain) during the period 1701-1850 based on documentary sources: evaluation and comparison with climate model simulations. Climate of the Past 8: 117-133. https://doi.org/10.5194/cp-8-117-2012, 2012.

A reconstruction from historical documents based on ordinal indices may be found in

Rodrigo, F.S., and Barriendos, M.: Reconstruction of seasonal and annual rainfall variability in the Iberian peninsula (16th-20th centuries) from documentary data. Global and Planetary Change 63: 243-257, doi: 10.1016/j.gloplacha.2007.09.004, 2008.

Thank you again for your comments,

F.S. Rodrigo
* * *

---

## Referee Comment (RC2) · Anonymous Referee #2 · 25 Feb 2019

The paper entitled "The Climate of Granada (southern Spain) during the first third of the 18th century (1706-1730) according to documentary sources" by Fernando S. Rodrigo deals with an interesting topic of historical climatology. It discloses and analyses new documentary (1706-1730) and instrumental (1728-1730) data from the south of the Iberian Peninsula and contributes to improve the knowledge of past climate in this part of Iberia The topic is no doubt within the scope of Climate of the Past. The author applies methods he has developed and published in his former works (Rodrigo, 2008; and Rodrigo, Gómez Navarro and Montávez-Gómez, 2012) to analyse newly found data. The presentation of the results is clear and well structured and the quality of figures and tables is very good. Although it is well written, I think the text should be

proof- read by a native English speaker.

Main points to review Data used. The author is revealing new, interesting and useful data. It is one more important piece to complete the puzzle of historical climatology in Iberia in the 18th century. However, some more details about the data are needed. p.2, l.32 – Describe the qualitative data more accurately. Which type of "summary"? Specify the "climatic conditions" referred to by the author p. 2, l. 33-35 – From your sentence beginning with "We establish" and ending with "Year", I would like to confirm that qualitative information only refers to certain years (as 1727 indicated on p. 3?). If so, this should be stated more clearly in the text. p.5, l.26-27 - Referring to instrumental data, couldn't you give more details? Or present a table as an example of one of Navarrete's? Or a facsimile as supplementary material? It is not indispensable but it would help the reader. Where in Granada were Navarrete's instruments placed?

Methodology I understand that the author does not want to repeat the statistical explanation in more detail, because the methodology has already been presented and discussed in former papers of the author (2008 and 2012). I agree with the author.

p.3, l.18 – . . . "The risk is to consider that these events were the 'normal' conditions. . ." I am afraid that I do not agree with this sentence. It is well known that in documentary data mostly extreme weather episodes are recorded and usual conditions are not mentioned, as C. Pfister wrote when he first presented this methodology (e.g. Pfister 1992, reference in Brázdil, 2005). That is why for certain years/seasons/months there are no data, as I understand happens with your documentary data. I suggest that you state more clearly the advantages of this methodology in relation to Pfister's indices.

Results The comparison of the average and standard deviation of the reconstructed period (1706-1730) to those of two periods of the 20th century of different temperature signal gives very interesting results, particularly for temperature where data from the reconstructed period are more similar to the beginning of the 20th century, a cooler period that occurred before the 20th century warming. p.5, l.7 – The "Maunder Minimum" ending date is 1715, so the period you are studying is only partly "subsequent to the cold Minimum Maunder". Suggestions: write "the coldest years of the Maunder Minimum in Central and Northern Europe". Do not forget your studied period includes very cold years, such as 1709. And if you look at your Table 1, most of the other cold seasons you have detected are not from the beginning of the period: they occur between 1723 and 1730 (except for summer 1710). Could the coldest period have ended later in Southern than in Northern Europe? In Portugal (Taborda et al, 2004) the two first decades of the 18th century were very cold. This could be discussed.

Discussion is missing. Either you include a "Discussion section" with a examination of results and comparison of your outcomes with other not only in Iberia, but also in Europe. See also former paragraph.

Or you drop the discussion and in this case it is advisable to develop the conclusion.

Conclusion Conclusion to be developed if discussion is not included.

Minor points abstract, l.8 – after 1706 indicate also the last year of the documentary data ( e.g. from 1706 to 1730) p. 1, l. 17 - after because include "they" p.1, l.18 – This sentence is unclear " ..using the methodological basis of historical climatology" p.1, l.22- Write "Alcoforado" unstead of "Alcaforado" p. 1, l.24 – review English formulation "in Spain normally has been considered" p.1, l.24 – What do you mean by "normally"? I think this word could be deleted because you have a reference at the end p.1, l. 29 – dated from instead of dated on p.1, l.29 - dated from instead of dated on p.1, line 33 – replace "since 1706" with "from 1706 to 1730" p. 2, l. 1 Indicate the dates of the beginning and the end of the Maunder Minimum (1645-1715) p.2, l.8 – "where he lived" instead of "where lived" p.2, l.9 and l.10 – "founded in" instead of "founded on" p.3, l.19 – What do you mean by "maintains this view"? p.3, l.20 – Refer that overlap period is essential not only to "validate the index" but to reconstruct long series of a climate variable (demonstrated by Brazdil et al., 2010, p. 16 and 17) p.4, l.23 – "applying the method". Explain p.4, l.37 – I suggest to delete global. p.7, l.28 – why this new

reference period? p.8, l.1 – Until 1715 it was still the Maunder minimum and in this paper you are not comparing the period 1706-30 with former periods, so it would be better to reformulate this sentence (see also note referring to p.5, l.7)

Tables 1 and 2 –indicate data sources Fig.2- Insert more information within caption (Granada station, studied period ..)

––––––––––––––––––––––––––––––––

---

## Author Comment (AC2) · 7 Mar 2019

Reply to Referee#2

Thank you very much for your comments and suggestions. They will be taken into account in the revised version of the manuscript.

In your Interactive Comment you include the following question: Could the coldest period have ended later in Southern than in Northern Europe? In Portugal (Taborda et al, 2004) the two first decades of the 18th century were very cold. This could be discussed.

[Figure]

Effectively, my results contrast with the analysis by Taborda et al (2004) on southern Portugal, where the two first decades of the 18th century were very cold. A possible explanation may be the variation of climate conditions from west to east in southern Iberian Peninsula. The climate of Granada is characterized by a diminishing of the Atlantic mechanisms that affect southwestern Iberian Peninsula, and strengthening influence of the Mediterranean mechanisms. The convenience of distinguish between western and eastern stations (particularly in winter) was highlighted in a previous work (Rodrigo, 2018). We must note that the period 1706-1730 is immediately subsequent to the coldest years of the Maunder Minimum in Central and Northern Europe. Luterbacher et al. (2004, 2007) and Xoplaki et al. (2005) found a warming trend in European winter and spring temperatures from the late Maunder Minimum, culminating in the late 1730s. On the other hand, the mean value of the autumn temperature in Central England between 1729 and 1738 was 10.5 ïĆřC, equal to that recorded during 1991-2000 (Jones and Briffa, 2006). Warming from the markedly cold decade of the 1690s to the 1730s is probably due to the scarcity of major explosive volcanic eruptions from the early 1700s compared to the previous two decades (Jones and Briffa, 2006). If there were differences between southern and northern Europe is an open question, but our results suggest that temperature trends in Granada were similar to those of central and northern Europe.

This comment will be included in the revised version of the manuscript, where a new section (Discussion) will be included, as you suggest.

Reference:

Rodrigo, F.S.: A review of the Little Ice Age in Andalusia (southern Spain): results and research challenges. Geographical Research Letters, 44: 245-265. doi: http://doi.org/10.18172/cig.3316, 2018.

Thank you again for your comments,

F.S. Rodrigo

---

## Author Response (AR1)

Dear Editor, in the following I explain the modifications of the manuscript cp-2018-170 entitled "The climate of Granada (southern Spain) during the first third of the 18th century (1706-1730) according to documentary sources" (reviewers' comments in bold).

Editor

**I would also highly appreciate it if you add one picture of Navarrete's manuscript in the source description.**

Done, I have included the new Figure 1 with the cover page of the book.

Referee #1

**1) The summary reflects the content of the article but should perhaps more clearly highlight its main contribution and novelty, i.e. applying a methodology alternative to Pfister-indices to a new and original set of documentary data.**

Done (page 1, line 11 of the revised version):

'A methodology alternative to Pfister-indices, based on the frequency of extreme events, was applied to study this new set of documentary data'.

**2) Documentary data are available for a period inferior to three decades. As climatological normals are used as baseline to evaluate climate events and provide context for year-to-year variability, is it a weakness for analysis and statistical comparisons?**

This question was answered in the reply during the open discussion. In the revised versión of the manuscript this question in briefly answered (page 4, lines 14-16):

'This methodology does not try to provide the year-to-year variability but the general characteristics of the studied period. This is a weakness of the analysis, although it is possible to reconstruct this inter-annual variability when documentary and instrumental periods are consecutive (Rodrigo et al., 2012)'

The length of the period (25 years, inferior to three decades) is not a serious problem. In fact, the last published report of the IPCC (IPCC, 2014: Climate Change 2014: Synthesis Report. Contribution of Working Groups I, II and III to the Fifth Assessment Report of the Intergovernmental Panel on Climate Change [Core Writing Team, R.K. Pachauri and L.A. Meyer (eds.)]. IPCC, Geneva, Switzerland, 151 pp) shows projections of average temperature and precipitation under different scenarios for 2081-2100 relative to the reference period 1986-2005, that is, using 20-years periods.

**3) Before using the methodology alternative to Pfister-indices, would it not be useful to establish the robustness of Navarrete's observations by comparing the indices drawn from his work with other available series?**

Done (page 3, lines 15-20):

'Some of these extreme seasons are confirmed by other data sources. So, for instance, cold winters 1709, 1729, and 1730 have been reported in other Spanish cities, as Tortosa, Seville, and Alicante (Alberola Romá, 2014), as well as the drought during the 1720s decade, in Jerez de la Frontera (AHVM, 1722), Arcos de la Frontera (ACAF, 1723), and Sevilla (Zúñiga, 1747), where *pro-pluvia* rogations were celebrated. According to Domínguez-Castro et al. (2010), droughts in Spain from early 18[th] century to 1730s are very scarce and their extension is very limited, except precisely in 1724, coinciding with the observations by Navarrete.'

**4) In the same logic, after application of the method proposed by Rodrigo (2008), would a cross comparison between the reconstruction made and another series (1906-1930 and 1976-2000) not be useful to strenghten the evidence?**

Done (page 5, lines 16-25, and Tables 4 and 5):

'Results are compared with the corresponding values of two 25-year periods in the 20[th] century, 1906-1930, and 1976-2000, when the warming signal is very different. To obtain a best view of this comparison, Table 5 shows the statistics corresponding to these periods. According to these results, seasonal mean temperatures during 1706-1730 were very similar to those during 1906-1930, even slightly warmer, and lesser (except in summer) than temperatures during 1976-2000, around 0.7 °C in winter, 0.4 °C in spring, and 1 °C in autumn. Standard deviations of temperature during 1706-1730 was similar to 1906-1930, and lesser than that of 1976-2000, suggesting smaller variability in the past. Total rainfall shows values very similar in autumn for the three periods, slightly wetter conditions in spring during 1706-1730 and 1906-1930, and slightly wetter conditions in winter of 1706-1730 in comparison with 1906-1930. Variability of rainfall in 1706-1730 is similar to that in 1976-2000, except in spring, characterized during 1976-2000 by drier conditions'.

**5) Is the observation program established by Navarrete in 1737 original and how does it fit into the cultural context of the time?**

This question is answered in page 2, lines 26-29:

'This program was based on the main ideas of the neo-Hippocratic hypothesis, which was the predominant medical paradigm during the 18[th] century in Spain. Unfortunately, this program was not accomplished due to the lack of interest of the authorities, although it was partially recovered at the last decades of the century by the medical academies of Seville, Madrid, and Barcelona (Anduaga Egaña, 2012).'

**6) - Page 2, line 25: "Precursor" or "Archetype" rather "A precedent"?**

Done (page 2, line 30)

**7) Page 4, line 28: "All the correlation coefficients were significant at the 95% confidence level." A statement to reformulate?**

Precisely, the statistical significance of correlation coefficients allows trust in the validity of the method, at least from a statistical point of view.

**8) Page 8, line 1: "a period of certain recovery", is the medical metaphor relevant? Perhaps "transition to a new phase after the cold Maunder Minimum period."?**

Done (page 9, lines 5-6).

**9) Page 8, lines 9-10: A stronger conclusion would be useful to highlight the contribution of the article on a poorly documented period for Spain?**

This comment has been included in the new section 5. Discussion (page 8, lines 2-5):

'In this work we have reconstructed the climatic mean conditions of a poorly documented period for Spain (the first third of the 18$^{th}$ century) in Granada (southern Spain) using documentary data. To date, there have been few attempts to reconstruct temperatures in the Iberian Peninsula, due to the scarcity of information (Bullón, 2008). Therefore, this work represents a new contribution to reconstruct historical temperatures in Spain'.

**10) Table 1: Addition of a temporal comparison with another space or very precises rogations series available (https://www.clim-past-discuss.net/cp-2018-67/ for exemple) is perhaps relevant?**

The manuscript quoted by the referee (Tejedor et al., 2018) is focused on the study of droughts in northeastern Spain, where climatic conditions are different to those of southern Spain. Tejedor et al. define an annual drought index, and show that the period 1706-1717 had low values of this index in northeast Iberian Peninsula, coinciding with the absence of dry seasons in Granada those years (Table 1). In any case, the manuscript by Tejedor et al. is now under review and I have preferred not use their results because they still can modify their methods and results.

Referee #2

**1)Although it is well written, I think the text should be proof- read by a native English speaker.**

The new versión of the manuscript has been revised by a native English speaker.

**2)some more details about the data are needed. p.2, l.32 – Describe the qualitative data more accurately. Which type of "summary"? Specify the "climatic conditions" referred to by the author p. 2, l. 33-35 . From your sentence beginning with "We establish" and ending with "Year", I would like to confirm that qualitative information only refers to certain years (as 1727 indicated on p. 3?). If so, this should be stated more clearly in the text.**

Done. In section 2 (page 2, lines 36-39) it is said that

'The chapter XVI is entitled 'Medical observations for the knowledge of climate'. Here, the author offers, at monthly and/or seasonal resolution, a summary of climatic conditions (rainfall, dryness, snowfalls, frosts, warm or cold weather, winds) from 1706 to 1730 in Granada, as well as their relationships with the appearance of illness in the city, following the neo-Hippocratic paradigm'.

And in section 3 (page 3, lines 4-9):

'The chapter XVI of the book by Navarrete (pages 105-107 of the manuscript) is dedicated to expose the 'alterations of health due to mutations of the air, and general causes obtained from the long observation and practical knowledge of the country'. Here, the author establishes relationships between different diseases and climatic conditions. Qualitative information only refers to certain years, when extreme events occurred. So, for instance, in the thirteen paragraph (page 106v), he says that "If cold, rain and snowfalls continue until May: difficult births, chest pains, and dangerous anginas: year 1727". This paragraph allows to characterize the spring of 1727 as wet and cold.'

**3)Referring to instrumental data, couldn't you give more details? Or present a table as an example of one of Navarrete's? Or a facsimile as supplementary material? It is not indispensable but it would help the reader. Where in Granada were Navarrete's instruments placed?**

More details are given in page 5, lines 33-40, including as example the information related to August 1729. Instead of adding supplementary material or a new table, and to avoid enlarge excessively the manuscript, the reader can access to the data file referenced:

'information is not presented tabulated, but as monthly summaries, indicating characteristic values or corresponding to critical moments, and not cover in detail all the days of the period. Sometimes, he adds comments on winds and other meteorological events (fog, cloudiness), and he indicates the number of rainy days of some months. So, for instance, for August 1729 he indicates that 'August began with warm weather, the day 2 the thermometer indicated 34, and a southern wind flew. Day 8 the thermometer increased two lines, from 38 to 40, during the total lunar eclipse, which was at one. Day 14 seemed the warmest day of the year, however the thermometer indicated 37, and from day 18 onwards there were slight northern winds, and the temperature decreased to 46'. This information was tabulated for analysis and may be found in Rodrigo (2018, NavarreteData.xlsx., page Gr1728-1730).'

In relation to instruments location, I indicate (page 6, lines 1-2) that

'There is no information about the installation of the instruments or the exact time at which readings were taken'

**4)p.3, l.18 – . . . "The risk is to consider that these events were the 'normal' conditions." I am afraid that I do not agree with this sentence. It is well known that in documentary data mostly extreme weather episodes are recorded and usual conditions are not mentioned, as C. Pfister wrote when he first presented this methodology (e.g. Pfister 1992, reference in Brázdil, 2005). That is why for certain years/seasons/months there are no data, as I understand happens with your documentary data. I suggest that you state more clearly the advantages of this methodology in relation to Pfister's indices.**

Done, page 4, lines 16-28, including a new reference (Rutherford et al., 1995):

'However, this methodology has advantages in comparison with the standard indices methodology. First, ordinal indices may be skewed by the subjectivity of the authors in original sources, and/or by the interpretation of the researcher of descriptions in the sources. In addition, ordinal indices are often based on the impact of the events on the socioeconomic infrastructures (for example, destruction of bridges during a river flood, loss of harvests, etc), and these impacts may change in different historical periods. The risk here is to consider as heavy extremes certain events that show the vulnerability of the system more than the extreme character of climate variables. The method followed is not based on the severity of the phenomena, and, in

consequence, at least to a certain degree, avoids these problems. In second place, it does not need an overlapping period with instrumental data, which is necessary to calibrate and validate indices, and to reconstruct a climate variable. There is a third problem of statistical nature: the calibration of indices normally is made using a regression procedure between proxy data (indices) and instrumental data during an overlapping period. From a statistical point of view, the consequence is the loss of variance of the reconstructed series, a problem that normally is solved using an 'inflation factor' to correct the reconstructed series (Rutherford et al., 1995). With this method, in principle, it is not necessary to introduce this mathematical artefact'.

**5)Results The comparison of the average and standard deviation of the reconstructed period (1706-1730) to those of two periods of the 20th century of different temperature signal gives very interesting results, particularly for temperature where data from the reconstructed period are more similar to the beginning of the 20th century, a cooler period that occurred before the 20th century warming. p.5, l.7 – The "Maunder Minimum" ending date is 1715, so the period you are studying is only partly "subsequent to the cold Minimum Maunder". Suggestions: write "the coldest years of the Maunder Minimum in Central and Northern Europe". Do not forget your studied period includes very cold years, such as 1709. And if you look at your Table 1, most of the other cold seasons you have detected are not from the beginning of the period: they occur between 1723 and 1730 (except for summer 1710). Could the coldest period have ended later in Southern than in Northern Europe? In Portugal (Taborda et al, 2004) the two first decades of the 18th century were very cold. This could be discussed.**

Done (page 8, lines 12-13):

'We must note that the period 1706-1730 is immediately subsequent to the coldest years of the Maunder Minimum in Central and Northern Europe.'

In relation to the discussion proposed by the referee, see the comments in page 8 lines 5-19, including new references (Taborda et al., 2004, suggested by the referee, and Rodrigo, 2018b):

'Results suggest that during 1706-1730 temperatures were very similar to those of the first decades of the 20$^{th}$ century, when the warming signal may be considered very small in comparison with the last decades of the 20$^{th}$ century. This result contrasts with the analysis by Taborda et al (2004) on southern Portugal, where the two first decades of the 18$^{th}$ century were very cold. A possible explanation may be the variation of climate conditions from west to east in southern Iberian Peninsula. The climate of Granada is characterized by a diminishing of the Atlantic mechanisms that affect southwestern Iberian Peninsula, and strengthening influence of the Mediterranean mechanisms. The convenience of distinguish between western and eastern stations (particularly in winter) was highlighted in a previous work (Rodrigo, 2018b). We must note that the period 1706-1730 is immediately subsequent to the coldest years of the Maunder Minimum in Central and Northern Europe. Luterbacher *et al*. (2004, 2007) and Xoplaki *et al*. (2005) found a warming trend in European winter and spring temperatures from the late Maunder Minimum, culminating in the late 1730s. On the other hand, the mean value of the autumn temperature in Central England between 1729 and 1738 was 10.5 °C, equal to that recorded during 1991-2000 (Jones and Briffa, 2006). Warming from the markedly cold decade of the 1690s to the 1730s is probably due to the scarcity of major explosive volcanic eruptions from the early 1700s compared to the previous two decades (Jones and Briffa, 2006). If there were differences between southern and northern Europe is an open question, but our results suggest that temperature trends in Granada were similar to those of central and northern Europe'.

**6)Discussion is missing. Either you include a "Discussion section" with a examination of results and comparison of your outcomes with other not only in Iberia, but also in Europe. See also former paragraph. Or you drop the discussion and in this case it is advisable to develop the conclusion. Conclusion. Conclusion to be developed if discussion is not included.**

5       Done. The manuscript has been rewritten including the new section 5. Discussion. (page 8).

**7)Minor points abstract, l.8 – after 1706 indicate also the last year of the documentary data ( e.g. from 1706 to 1730) p. 1, l. 17 - after because include "they" p.1, l.18 – This sentence is unclear " ..using the methodological basis of historical climatology" p.1, l.22- Write "Alcoforado" unstead of "Alcaforado" p. 1, l.24 – review English formulation "in Spain**
10       **normally has been considered" p.1, l.24 – What do you mean by "normally"? I think this word could be deleted because you have a reference at the end p.1, l. 29 – dated from instead of dated on p.1, l.29 - dated from instead of dated on p.1, line 33 – replace "since 1706" with "from 1706 to 1730" p. 2, l. 1 Indicate the dates of the beginning and the end of the Maunder Minimum (1645-1715) p.2, l.8 – "where he lived" instead of "where lived" p.2, l.9 and l.10 – "founded in" instead of "founded on" p.3, l.19 – What do you mean by "maintains this view"? p.3, l.20 – Refer that overlap period**
15       **is essential not only to "validate the index" but to reconstruct long series of a climate variable (demonstrated by Brazdil et al., 2010, p. 16 and 17) p.4, l.23 – "applying the method". Explain p.4, l.37 – I suggest to delete global. p.7, l.28 – why this new reference period? p.8, l.1 – Until 1715 it was still the Maunder minimum and in this paper you are not comparing the period 1706-30 with former periods, so it would be better to reformulate this sentence (see also note referring to p.5, l.7)**

Suggestions and typo corrections indicated by the referee have been included in the new version of the manuscript. Unclear sentences have been clarified. In the study of rainy days it was used the reference period 1971-2000 because (page 7, lines 32-33):

25       'data on rainy days are not available in the database by Luna et al (2005), therefore, we used the AEMET climate summary of the reference period 1971-2000'

**8)Tables 1 and 2 –indicate data sources Fig.2- Insert more information within caption (Granada station, studied period**

30       Done.

I wish express my gratitude to the anonymous referees by their useful comments and suggestions. I hope that the manuscript be able for publishing in 'Climate of the past'.
Sincerely,
35       F.S. Rodrigo

[revised manuscript text omitted]